# Deciphering Microbial Composition in Patients with Inflammatory Bowel Disease: Implications for Therapeutic Response to Biologic Agents

**DOI:** 10.3390/microorganisms12071260

**Published:** 2024-06-21

**Authors:** Orazio Palmieri, Fabrizio Bossa, Stefano Castellana, Tiziana Latiano, Sonia Carparelli, Giuseppina Martino, Manuel Mangoni, Giuseppe Corritore, Marianna Nardella, Maria Guerra, Giuseppe Biscaglia, Francesco Perri, Tommaso Mazza, Anna Latiano

**Affiliations:** 1Division of Gastroenterology and Endoscopy, Fondazione IRCCS “Casa Sollievo della Sofferenza”, 71013 San Giovanni Rotondo, Italy; f.bossa@operapadrepio.it (F.B.); tiziana.latiano@gmail.com (T.L.); s.carparelli@operapadrepio.it (S.C.); g.martino@operapadrepio.it (G.M.); giuseppe.corritore@gmail.com (G.C.); mariaguerra.mg247@gmail.com (M.G.); giuseppe.biscaglia@gmail.com (G.B.); f.perri@operapadrepio.it (F.P.); a.latiano@operapadrepio.it (A.L.); 2Unit of Bioinformatics, Fondazione IRCCS “Casa Sollievo della Sofferenza”, 71013 San Giovanni Rotondo, Italymanulel.mangoni@uniroma1.it (M.M.); t.mazza@operapadrepio.it (T.M.); 3Department of Experimental Medicine, Sapienza University of Rome, 00161 Rome, Italy

**Keywords:** biologic drugs, CD, microbiome, UC

## Abstract

Growing evidence suggests that alterations in the gut microbiome impact the development of inflammatory bowel diseases (IBDs), including Crohn’s disease (CD) and ulcerative colitis (UC). Although IBD often requires the use of immunosuppressant drugs and biologic therapies to facilitate clinical remission and mucosal healing, some patients do not benefit from these drugs, and the reasons for this remain poorly understood. Despite advancements, there is still a need to develop biomarkers to help predict prognosis and guide treatment decisions. The aim of this study was to investigate the gut microbiome of IBD patients using biologics to identify microbial signatures associated with responses, following standard accepted criteria. Microbiomes in 66 stool samples from 39 IBD patients, comprising 20 CD and 19 UC patients starting biologic therapies, and 29 samples from healthy controls (HCs) were prospectively analyzed via NGS and an ensemble of metagenomics analysis tools. At baseline, differences were observed in alpha and beta metrics among patients with CD, UC and HC, as well as between the CD and UC groups. The degree of dysbiosis was more pronounced in CD patients, and those with dysbiosis exhibited a limited response to biological drugs. Pairwise differential abundance analyses revealed an increasing trend in the abundance of an unannotated genus from the *Clostridiales* order, *Gemmiger* genus and an unannotated genus from the *Rikenellaceae* family, which were consistently identified in greater abundance in HC. The *Clostridium* genus was more abundant in CD patients. At baseline, a greater abundance of the *Odoribacter* and *Ruminococcus* genera was found in IBD patients who responded to biologics at 14 weeks, whereas a genus identified as *SMB53* was more enriched at 52 weeks. The *Collinsella* genus showed a higher prevalence among non-responder IBD patients. Additionally, a greater abundance of an unclassified genus from the *Barnesiellaceae* family and one from *Lachnospiraceae* was observed in IBD patients responding to Vedolizumab at 14 weeks. Our analyses showed global microbial diversity, mainly in CD. This indicated the absence or depletion of key taxa responsible for producing short-chain fatty acids (SCFAs). We also identified an abundance of pathobiont microbes in IBD patients at baseline, particularly in non-responders to biologic therapies. Furthermore, specific bacteria-producing SCFAs were abundant in patients responding to biologics and in those responding to Vedolizumab.

## 1. Introduction

Inflammatory bowel disease (IBD) is characterized by non-infectious chronic inflammation of the gastrointestinal tract, and the most common forms include Crohn’s disease (CD) and ulcerative colitis (UC). They are of high medical and socioeconomic relevance, with around 6–8 million cases of IBD worldwide, and their prevalence is rising in industrialized countries [1]. More than 200 genetic risk factors for the development of IBD have been identified [2,3], but none of them have clear roles as biomarkers. Mucosal healing, characterized by the regression or disappearance of endoscopic lesions, is a primary treatment goal in moderate and severe disease courses; here, the aim is to achieve and maintain clinical remission or mucosal healing [4], often requiring the use of immunosuppressant drugs and molecules with diverse mechanisms of action, including the inhibition of leukocyte trafficking (anti-integrins) or blocking the effect of inflammatory cytokines such as TNF-alpha and IL12/23-inhibitors [5], collectively called “biologic therapies”.

However, despite the expansion of the therapeutic armamentarium [6], a proportion of patients do not achieve or maintain disease remission. To date, the reasons why some patients fail to respond to a particular drug remain poorly understood. 

Prognostic factors have been extensively studied. They include clinical, endoscopic, radiologic, genetic, proteomic, transcriptomic, serological and microbial factors [7].

In clinical practice, although there is evidence that markers such as Oncostatin M [8], IL13RA2 [9] and TREM-1 [10] might predict a response to biologic therapy, they cannot yet be applied at the bedside. 

The limited predictive value of biomarkers for therapeutic response makes it difficult to determine a priori the most beneficial drug for a specific patient. Consequently, there is growing interest in developing a therapeutic pipeline and identifying predictive biomarkers for a response to a particular therapy. 

Reports on animal models [11], association studies [12] and analyses of alterations in the gut microbiome [13] suggest that host–microbe interaction is often required for genetically susceptible individuals to develop colitis. This supports the notion that the intestinal microbiome plays a central role in the pathogenesis of both CD and UC. Recently, several studies [14,15,16,17] on IBD patients starting a course of biologic therapy have used microbial compositions to predict responses to these biological molecules. They identified a significant presence of taxa with anti-inflammatory effects, butyrate-producing microbial species, mucin-degrading bacteria and species capable of producing short-chain fatty acids (SCFA), especially in specimens from healthy controls (HCs). However, it is unclear which populations of microbes are involved or how they might contribute to IBD or help the host’s response to therapy.

Despite advancements, biomarkers that can predict prognosis and response and that can guide stratified therapeutic approaches in relation to IBD are still an unmet medical need. 

In this study, we aimed to identify microbial signatures that are predictive of treatment response.

## 2. Materials and Methods

This was a prospective, single-center study. The inclusion criteria were as follows:(1)Patients with an established diagnosis of IBD (CD or UC) who had started biologic therapy between June 2018 and August 2020.(2)Patients with active disease as assessed by clinical indices (namely, a Harvey–Bradshaw Index (HBI) score ≥5 and a Mayo partial score >2) and at least one of the following objective markers of disease activity: a C-reactive protein (CRP) concentration >5 mg/L, a fecal calprotectin level >250 µg/g and endoscopic or radiological signs of activity.

Response to therapy was assessed using clinical indices (HBI < 5 and PMS < 2) and at least one objective marker of disease activity (a CRP level <5 mg/dL, a fecal calprotectin level <150 µg/g, a reduction of at least 3 points in SES-CD or ≥1 point in endoscopic Mayo score, or a reduction of at least 2 mm in bowel thickness as assessed by bowel ultrasound or small-bowel MR enterography).

A total of 66 stool samples from 39 IBD patients (20 with CD and 19 with UC) and 29 HCs were prospectively collected at the Gastroenterology and Digestive Endoscopy Unit at Fondazione IRCCS-Casa Sollievo della Sofferenza Hospital. From these IBD patients starting biologic therapies, 39, 11 and 16 specimens were collected at baseline, 14 weeks and 52 weeks, respectively. Fecal samples from both IBD and HC subjects were collected at their homes and immediately frozen inside a sterile container at −20 °C and subsequently stored at −80 °C at the hospital’s research laboratory. 

This study was conducted in accordance with the Declaration of Helsinki, and the protocol was approved by the hospital’s ethics committee (Prot. N.132 CE/2015). All subjects provided signed informed consent forms. 

The control group consisted of healthy, non-IBD-afflicted, non-hospitalized individuals (HCs—healthy controls). The HCs were subjects recruited from among laboratory personnel and stool donors for fecal microbiota transplantation. None of the HCs had a history of gastrointestinal disease or malignancy or used probiotics, antibiotics or supplements capable of altering their microbial compositions for at least 3 months before sampling.

### 2.1. Assessment of Disease Activity

To evaluate changes in disease activity in IBD patients, assessments were conducted at visit 1 (baseline), which constituted the visit before the initiation of the biologic treatment; visit 2 (14 weeks); and visit 3 (52 weeks). For UC patients, disease activity was assessed using the partial Mayo score [18], whereas CD was assessed using the Harvey–Bradshaw Index (HBI) [19], and the site of disease was defined according to the Montreal classification [20]. We considered responders to specific biologics as those patients showing a reduction of ≥2 points in PMS and ≥3 points in HBI from baseline, associated with at least one of the following objective markers of inflammation: a CRP level <5 mg/dL, a fecal calprotectin level <150 µg/g, a reduction of at least 3 points in SES-CD or ≥1 point in the endoscopic Mayo score, or a reduction of at least 2 mm in bowel thickness as assessed by bowel ultrasound or small-bowel MR enterography.

### 2.2. Laboratory Procedures 

DNA was extracted by using the QIAamp PowerFecal Kit (Qiagen, Hilden, Germany) according to the manufacturer’s recommendations. DNA quantity was examined using a NanoDrop ND-1000 spectrophotometer (Thermo Fisher Scientific, Inc., Somerset, NJ, USA). Microbial diversity analysis of the mucosal specimens was conducted by sequencing the amplified V3 to V4 hypervariable region of the 16S rRNA gene. PCR primers and conditions were employed and set, respectively, according to the Illumina 16S Metagenomic Sequencing Library preparation guide (Part # 15044223 Rev.B) [21] with the following exceptions: for the first 16S PCR, the process was performed using Taq Phusion High-Fidelity (Thermo Fisher Scientific, Sunnyvale, CA, USA) in 25 µL reaction volumes, and 25 cycles were used in the PCR.

Subsequently, the amplicons were purified using AMPure XP beads (Beckman Coulter, Milan, Italy). Afterward, the ligation of the dual indexing adapters was performed in the presence of Nextera XT Index Primer 1 and Primer 2 (Illumina, San Diego, CA, USA), Taq Phusion High-Fidelity (Thermo Fisher Scientific) and 5 μL of purified DNA, according to the manufacturer’s instructions. The products were purified using AMPure XP beads to create the final cDNA library. Library concentrations and fragment sizes were measured using a fluorometric-based system (Qubit dsDNA BR Assay System; Thermo Fisher Scientific) and an Agilent 2200 TapeStation Bioanalyzer (HS D1000 ScreenTape Assays; Agilent Technologies, Santa Clara, CA, USA), respectively. Equal amounts of cDNA libraries were pooled, denatured with NaOH, diluted with hybridization buffer to 7 pM in accordance with Illumina’s protocol and spiked with 20% PhiX (Illumina). The libraries were loaded into a flow cell V2 (500 cycles) via paired-end sequencing (2 × 250) (Illumina) and sequenced with the MiSeq sequencing instrument system (Illumina) according to the manufacturer’s recommendations. 

### 2.3. Bioinformatics and Statistical Data Analysis

The quality of raw reads was primarily checked through the FastQC application, v. 0.12.0. The QIIME2 v.2021.11 suite [22] was employed for the primary analysis, which included read quality checking and filtering, de-replication, chimeric read identification and paired-end read joining. Sequence similarity clustering was performed by employing the QIIME’s DADA2 [23] plugin; a raw feature table was generated. The taxonomic nomenclature for the detected features was inferred using the QIIME2 embedded Naïve Bayes fitted classifier, pre-trained on the Greengenes reference database v.13.8 [24]. Rarefaction curve analysis via the QIIME2 diversity plugin was used to estimate the completeness of microbial community sampling, using different sampling depth thresholds (5000, 10,000 and 20,000 reads per sample). Subsequently, alpha indices (Shannon’s diversity index [25], number of observed OTUs, Faith’s phylogenetic diversity [26] and Pielou’s evenness [27]) and beta dissimilarities (Jaccard distance [28], Bray–Curtis distance [29] and unweighted and weighted UniFrac distances [30]) were calculated at the sample level. Principal coordinates analysis (“PCoA”) plots were developed using the EMPeror application [31] for beta diversity metrics to check intra- and inter-group heterogeneity. The Kruskal–Wallis and PERMANOVA statistical tests (number of permutations = 9999) were applied to detect differences regarding the above-mentioned alpha index and beta measures among sample groups, respectively. 

Feature abundance profiles among groups were investigated using compositional data analysis methods, i.e., the ANCOM (QIIME2 plugin), Selbal [32], Coda-lasso [33] and Clr-lasso [34] R Packages (https://malucalle.github.io/Microbiome-Variable-Selection, accessed on 8 December 2023). The raw genus-collapsed feature table was refined by removing non-bacterial sequences (i.e., those of mitochondrial/chloroplast origin) and “ultra-rare” taxa, i.e., those appearing in fewer than 5 samples out of the total 66 considered samples or counted less than 20 times across all samples. 

Together with the ANCOM method, the Coda-lasso, Clr-lasso and Selbal feature selection algorithms were additionally used, to detect a microbial signature within pairwise group comparisons (IBD, CD, UC, IBD 14 Weeks, IBD 52 Weeks, CD 14 Weeks, CD 52 Weeks, UC 14 Weeks, UC 52 Weeks, UC Vedolizumab 14 Weeks, UC Vedolizumab 52 Weeks, CD Vedolizumab 14 Weeks, CD Vedolizumab 52 Weeks, and HCs). Appendix A contains details on the group comparisons and methodological settings for each considered method. 

Furthermore, the inter-sample Bray–Curtis (BC) beta diversity metrics from QIIME2 analyses were used in the execution of the Lloyd–Price test [35] for dysbiosis detection. Briefly, a BC threshold was calculated as the 90% quantile of the pairwise BC distribution within the HC sample group. Thus, a median BC value was determined by comparing an IBD sample with each sample in the control group; IBD samples with a median BC > BC threshold were defined “dysbiotic”. 

Finally, cross-correlation among the genera abundance profiles among the IBD, HC, UC and CD sample groups was investigated by using the FastSpar v0.1 tool [36]. Feature counts were transformed as log ratios to be statistically treated as compositional data, as described by Friedman and Aim [37]. Empirical p-values for cross-correlation among real abundance profiles were obtained by calculating correlations for n = 1000 permutations of the real feature count table. 

Microbial networks at the genus level were built, considering genera (features) as nodes and the strength of the correlations of their abundance as edges (links). These networks were analyzed using Pyntacle v1.3 [38] to assess their density and identify important sets of genera. The density of the networks, defined as the proportion of edges over the size of a network, was assessed through the completeness index [39]. Subsequently, the density level of each investigated network was compared to three equitized Erdős–Rényi random network models that were generated with increasing wiring probability (*p* = 0.25, 0.5, and 0.7).

The topological importance of individual nodes was determined through the betweenness centrality local metric, which measures the extent to which a node is involved in pathways between nodes in a network. Sets of important nodes were identified through the keyplayers metrics, which give insights into the contributions of nodes to the cohesiveness of a network (F—fragmentation, whose values are between 1 and 0) and focus on the connectivity/embedding of nodes (m—reach, denoting reachability) within a network [40]. This analysis was conducted on sets of nodes <4 in size.

## 3. Results

### 3.1. Clinicopathologic Characteristics and Diversity Analysis

A total of 95 stool samples were collected, comprising 39 from IBD patients and 29 from HCs. The makeup of the patient cohort was as follows: 13 (33.3%) with moderate-to-severe IBD starting anti-TNF therapy (of which 6 had UC), 25 (64.1%) beginning Vedolizumab treatment (with 13 having UC) and 1 (2.6%) with CD starting Ustekinumab therapy. The mean age of the IBD patients was 34.5, and the mean duration of disease was 12.3 years. Among the UC patients, 2 had proctitis, 10 had distal colitis, and 7 had pancolitis. Disease localization in the CD patients was as follows: 13 had ileitis, 1 had colitis, and 6 had ileocolitis. At baseline, the mean values of the Mayo score and HBI were 9.0 and 8.5, respectively. The participants’ characteristics are given in Table 1. The control group consisted of 29 HCs, of whom 13 were female, with a mean age at recruitment of 46.1 ± 8.1 years. 

### 3.2. 16S rRNA V3-V4 Region Sequencing

After performing the QIIME2 quality control procedure, we obtained sample-specific sequencing yields ranging from 26,571 to 251,305 high-quality reads. Using the raw feature table and its corresponding phylogenetic tree, we conducted several rarefaction tests at different read depth cutoffs (refer to Appendix A). This allowed us to determine an optimal sampling depth with minimal sample loss. A rarefaction depth of 20,000 reads ensured a stable distribution of the alpha diversity metrics across all the investigated groups.

### 3.3. Microbial Diversity and Community Analyses 

When the QIIME pipeline was run, we observed differences in both alpha and beta diversities among the HCs and patient groups. In detail, alpha diversity was evaluated by analyzing four metrics (Shannon Entropy, Pielou’s evenness, number of observed features and Faith’s phylogenetic distance) and comparing group-specific distributions through Kruskal–Wallis tests, both global and pairwise. Table 2 reports q-values derived from these pairwise comparisons, and representations of the alpha and beta diversity metrics are shown in Appendix A.

When comparing the microbiota from IBD patients to those of the HC group, we highlighted that the microbial communities in IBD patients consistently had lower diversity in all the analyzed alpha indices (q values < 0.0019). This reduced diversity was especially evident in the CD cohort, wherein all metrics showed q values < 0.00014. On the other hand, the differences in the UC group showed a weak association (q values < 0.015), as reported for the Bray–Curtis index in Figure 1, and no significant differences were exhibited according to Pielou’s evenness index. 

In a direct comparison between the microbial communities of the CD and UC cohorts, the CD group displayed lower diversity across all alpha indices (q values < 0.013).

We compared the microbial samples using four beta dissimilarity metrics (Bray–Curtis, Jaccard, Unweighted Unifrac and Weighted Unifrac) through the PERMANOVA pairwise test, and the sample dissimilarities were visualized using Emperor Plots (see Appendix A).

The comparison of microbial compositions between the IBD and HC groups revealed statistically significant differences across all the analyzed metrics (q-value < 0.0018). When comparing both the CD and UC groups with the HCs, the differences remained statistically significant (with a q-value < 0.0012 for the CD patients and a q-value < 0.049 for the UC patients).

In a direct comparison of the microbiota of the CD and UC patients, significant differences were observed in the Bray–Curtis (Figure 2), Jaccard and Unweighted Unifrac metrics (q-value < 0.0067). However, the differences were not significant for the Weighted Unifrac metric, as detailed in Table 3.

We used the QIIME2 ANCOM module to assess the microbial compositions of the samples and identify significant variations among them. After applying a filtering process, feature tables measuring 108 × 68 (genus-collapsed features by samples) were used as input for the ANCOM analysis.

At the genus level (L6), when comparing the HCs to the IBD patients, several taxa exhibited differential abundance. These included the *Gemmiger* genus (ANCOM W statistics = 82), an unannotated genus from the *Rikenellaceae* family (W = 77), an unannotated genus from the *Clostridiales* order (W = 76) and the *Alistipes* genus (W = 73), all of which were significantly more abundant in the HC group.

In the comparisons between the CD, UC and HC groups at the genus (L6) level, it was revealed that the *Clostridium* genus (W = 104) was more abundant in the CD cohort.

### 3.4. Microbial Ecosystem Analysis

The degree of dysbiosis among the samples is shown in Figure 3 and Appendix A, displaying data on all the IBD patients at baseline and the HC samples. According to the results of the Lloyd–Price test, 14 of the 39 IBD patients (35.9%) showed a greater degree of dysbiosis than the control group (dysbiosis cut-off = 0.92). Of these patients, 10 (50%) were diagnosed with CD, and the remaining 4 had UC (21%). Among the dysbiotic patients, 5 of 14 IBD patients (35.7%) not responding to Vedolizumab after 52 weeks exhibited more pronounced dysbiosis, and none of the 11 responders exhibited dysbiosis (*p* = 0.046, Fisher’s Exact Test). In examining factors associated with the response to Vedolizumab, binary logistic regression analysis revealed that the median Bray–Curtis beta diversity metrics made a prediction of the response to Vedolizumab that approached statistical significance (OR = 1.20, 95% CI [0.99, 1.46], *p* = 0.066). This suggests that, for each 1% increase in Bray–Curtis beta diversity, the odds of a response to Vedolizumab increase by 20%. Given the close proximity to conventional levels of significance, it is recommended that studies with larger sample sizes or more sensitive measures are conducted to further explore this potential relationship.

### 3.5. Cross-Correlation Analysis

Microbial networks were constructed using 16S rRNA data. To assess co-occurrence across all study groups (i.e., IBD, CD, UC and HC), we computed pairwise Pearson correlation coefficients of microbial abundances and built a co-occurrence network for each study. Correlation coefficients >0.5 or <−0.5 and empirical *p*-values ≤ 0.05 are shown in the “Cross-correlation” sheet in Appendix A. For the UC cohort, we identified 138 co-occurring nodes (91 positive correlation values), 93 nodes (64 positive) in CD patients, 25 nodes (13 positive) in IBD patients and 137 nodes in HC patients (70 positive). Globally, all four networks were sparse, even compared with random networks of the same size and with an increasing number of edges (Table 4). This implies that genera were not greatly correlated in any of the four networks.

Three pairs of nodes were common between the CD patients and HCs. Pair 35–38 (these numbers are custom feature identifiers reported in Appendix A) had a positive co-occurrence value, and the other two pairs, 13–63 and 37–70, displayed opposite correlation coefficients. 

Seven pairs were shared between the UC and HC groups. Of these, three pairs (11–12, 19–75 and 62–63) had negative correlation values. Pair 11–13 was positively correlated. Additionally, pairs 12–45 and 12–63 were positive for the UC group but negative for the HCs, whereas pair 13–51 was negative for the UC group and positive for the HCs.

Seven pairs were common between the CD and UC groups. Of these, six exhibited positive correlation values, specifically pairs 13–32, 26–32, 33–73, 44–64, 49–67 and 59–90, whereas pair 6–21 showed a negative correlation value. Pair 12–13 had a negative correlation value for the CD, UC and HC groups.

We then measured the individual importance of genera using the betweenness centrality and keyplayers metrics. As for betweenness, in the HC network, genus 63 exhibited the highest value (143), so it was the most important node in the network. It was also among the top three in the UC group (with a score of 485), and genus 35 ranked first (with a score of 511). Genus 32 was the most central node in the CD network, with a score of 861 (compared to the second-best value of 772 for feature 55). Finally, for IBD, features 13 and 31 ranked the highest for betweenness, with scores of 54 and 45, respectively, with feature 65 ranking third with a score of 25 (Appendix A; sheet: “Betweenness”).

Given the low dimensionality of the networks in terms of both nodes (features) and edges (links), the keyplayer analysis was exploited to identify both individual (k = 1) and sets of nodes (k = 2 and k = 3) whose removal resulted in the maximal breakdown of the network or that were maximally reachable from the other nodes. In the IBD network, 13 was the feature whose removal compromised the integrity of the network (F = 0.8), and 31 was the best at reaching the rest of the network nodes with few links (m-reach = 11). Both proved to be important roles when considered in the group as well. The removal of 13 and 19 caused, in fact, the maximum amount of fragmentation, with a score of 0.86. Features 31 and 59, instead, exhibited the highest reachability score that was 13. Removing genus 65 together with genera 13 and 19 from the network caused an increase in fragmentation, with a score of 0.91, along with various features, combined with 31–59 having an m-reach = 14. This suggests a truly important role for features 13 and 31 in the IBD network of co-occurrences.

Moving on to the HCs, the main features that were constant in the keyplayer analysis were 63, which exhibited a fragmentation score of 0.61, and 11, which resulted in being quickly reachable by 35 other features. Increasing the size of the k-set did not particularly affect the cohesiveness of the network; thus, a fragmentation score of 0.67 was reached when removing features 63, 37 and 6. Nodes 11, 15 and 20 obtained the best reachability score, namely 39.

For the UC network, considering fragmentation, feature 35 alone achieved a score of 0.46, which increased to 0.63 in combination with 25 and 62. In terms of reachability, features 12, 30 and 27 were equivalent. However, only feature 30 could increase the number of other reached nodes to 43, i.e., more than half of the whole network, if coupled with one or two other nodes, i.e., 62 and 90.

Finally, for CD, feature 26 produced the highest fragmentation (0.58), and feature 55 could reach 22 other nodes. Considering groups of nodes with a size of 2, 32 and 55 exhibited the highest fragmentation scores, and 55 and 56 had the best reachability scores. Features 32, 55 and 70 exhibited a fragmentation score of 0.76, and features 55, 56 and 60 could reach 36 other features.

### 3.6. Discriminant Taxa from Pairwise Group Comparisons

We investigated differential microbial abundance between group pairs using the Coda-lasso, Clr-lasso and Selbal methods. Comprehensive results can be found in Appendix A.

A given feature “X” was deemed “common” if it was identified as “significantly differentially abundant” by all approaches (although weights are calculated differently among the methods). We found that four “genus-collapsed” taxa were consistently observable across at least three methods when comparing the HC cohort to the IBD samples. In the “HC vs. IBD” pairwise analysis, we highlighted that *Gemmiger* (9), an unannotated genus from the *Rikenellaceae* family (34), an unannotated genus from the *Clostridiales* order (23) and *Bilophila* (81) were more abundant in the HCs. It is noteworthy that genera identified with taxa 9, 34 and 23 were also commonly found using the ANCOM test. On the other hand, two taxa, *cc_115* from the *Erysipelotrichaceae* family (11) and *Meganomonas* (87), were more abundant in the IBD group.

When comparing the microbial composition of the HCs to that of the CD cohort, it was found that an unannotated genus from the *Clostridiales* order (23) was more prevalent in the HC group.

In the pairwise comparison of “HC vs. UC”, we observed that *Alistipes* (2), an unannotated genus from the *Rikenellaceae* family (34), and *Ruminococcus* (52) were more abundant in the HCs. Conversely, *cc_115* (11) was more represented in the UC cohort.

For the “CD vs. UC” comparison, *Streptococcus* (8), *Bacteroides* (12) and *Turicibacter* (26) were more prevalent in the UC group at the genus level. In contrast, in the CD group, two taxa, *Clostridium* (58) and *Ruminococcus* (52), were more dominant.

Both the feature selection algorithms and the ANCOM test highlighted the prominent presence of unique taxa dwelling primarily in the stool samples of HCs, namely *Gemmiger* (9), an unannotated genus from the *Rikenellaceae* family (34), and an unannotated genus from the *Clostridiales* order (23), with the *Clostridium* genus being mainly associated with CD.

### 3.7. Microbial Composition at Baseline, 14 Weeks and 52 Weeks

Only a minor difference in the alpha for Faith’s phylogenetic distance was observed when comparing the microbial compositions of all 39 IBD subjects to those of the 16 patients at 52 weeks, for whom specimens were available (q value = 0.06; *p* value = 0.04). The microbial composition at 52 weeks increased the alpha of Faith’s phylogenetic distance of gut microbiota. 

In the pairwise analysis using the Coda-lasso, Clr-lasso and Selbal methods, the *Enterococcus* (39) genus was found to be more abundant and associated with IBD patients at baseline when comparing the microbial compositions from baseline to those at 52 weeks (data are shown Appendix A).

### 3.8. Baseline Microbial Composition as a Predictor for Response to Biologic Therapy

Of the 39 IBD patients treated with biologics, 22 achieved a response at 14 weeks. Their baseline microbial compositions were compared to those of the 16 non-responders using the Coda-lasso, Clr-lasso and Selbal methods. At the “genus-collapsed” level, the patients responding to biologics at 14 weeks exhibited a higher abundance of *Odoribacter* (29) and *Ruminococcus* (52) at baseline. In contrast, *Collinsella* (49) was more prevalent among non-responders. At 52 weeks, a genus identified as *SMB53* from the *Clostridiaceae* family (73) was more enriched in the responders.

A sub-analysis was conducted to discern differences based on the specific drug used. Of the IBD patients, 25 were treated with Vedolizumab (13 UC and 12 CD patients), and 13 were treated with anti-TNF (7 CD and 6 UC patients). A total of 14 IBD patients (56%) who responded to Vedolizumab at 14 weeks had a greater abundance of taxa from the *Lachnospiraceae* (18) and *Barnesiellaceae* (75) families. However, at week 54, no significant associations were observed across all three methods. For UC patients on Vedolizumab at 14 weeks, the unannotated genus from the *Barnesiellaceae* family (75) was more abundant in responders, whereas the *Collinsella* genus (49) was more associated with non-responders. Additionally, by 52 weeks, the genus *cc_115* from the *Erysipelotrichaceae* family (11) was more prevalent among non-responding UC patients.

For CD patients responding to Vedolizumab at 14 weeks, there was an enrichment of an unannotated genus from the *Lachnospiraceae* family (68), whereas at 52 weeks, these responders showed a greater abundance of *Anaerotruncus* (68) and *Lactococcus* (72).

Due to the limited number of subjects receiving anti-TNF therapy, no significant associations were identified in any of the comparisons. 

## 4. Discussion

Growing evidence indicates that gut microbiome dysbiosis, interacting with genetic susceptibility, significantly impacts IBD development by reducing anti-inflammatory activity and disrupting various metabolic and regulatory functions.

Our analysis revealed a pronounced degree of dysbiosis in IBD patients, particularly those with CD, as indicated by both alpha and beta metrics, along with the Lloyd–Price test. Our data demonstrate that dysbiotic patients exhibited a limited response to biologic drugs. Further assessments conducted utilizing compositional data analysis algorithms (such as ANCOM, Coda-lasso, Crl-lasso and Selbal) unveiled a lower prevalence of certain bacterial taxa in IBD patients, whereas the *Clostridium* genus was more abundant in those with CD.

The analysis of microbiome compositions strongly indicated consistent identification of an unannotated genus from the *Clostridiales* order, the *Gemmiger* genus and an unannotated genus from the *Rikenellaceae* family using all methods. These genera were found in greater abundance in HCs than in the IBD group.

The unannotated genus from the *Clostridiales* order consistently exhibited a stronger association with the HC group than with the CD group. Similarly, an unannotated genus from the *Rikenellaceae* family was more frequently found in the HC group than the UC group. Additionally, when tested using the ANCOM method, the *Alistipes* genus demonstrated greater abundance in the HC group than in individuals with IBD. This observation was further confirmed in the pairwise analysis, showing that *Alistipes* was more prevalent in the HC group than in the UC group. 

The *Clostridiales* order, which includes well-known defensive symbionts such as *Faecalibacterium*, *Roseburia*, *Blautia* and *Gemmiger*, is recognized for its ability to suppress proinflammatory bacteria, produce SCFAs and induce an immune response [41]. However, a study conducted on a Chinese IBD cohort revealed a significant decrease in *Clostridiales* among patients with severe CD [42]. 

The *Gemmiger* genus was previously found to be more abundant in the stool samples of an HC group compared to those from individuals with CD [43]. This observation is consistent with our group’s findings [44], wherein *Gemmiger* was detected in higher concentrations in both colonic and ileal mucosa from healthy tissues compared to normal pouch mucosa or pouchitis samples from IBD patients. 

The *Rikenellaceae* family, which includes the *Alistipes* genus, among others, is known for producing butyrate. Our results indicate that there was a significant reduction in *Rikenellaceae* in the IBD patients, especially those with UC. This observation has been reported in both CD and UC patients [45,46]. Additionally, an increase in the abundance of *Alistipes* is associated with the mitigation of weight loss and the restoration of histological damage in mouse-induced colitis by Ornithine α-Ketoglutarate [47].

More importantly, the analysis of microbiome compositions revealed a pair of pathobiont microbes: the *Clostridium* genus, found in greater abundance in IBD patients, especially those with CD, and *cc_115* from the *Erysipelotrichaceae* family, which was more abundant in both IBD and UC. The *Clostridium* genus includes significant human and animal pathogens, causative of potentially harmful diseases [48], and has been found in greater abundance in the intestinal strictures of CD patients [49]. Although the evidence regarding changes in the levels of *Erysipelotrichaceae* does not appear consistent, previous observations in patients with IBD or animal models of IBD have reported such associations [50].

The FastSpar cross-correlation analysis depicted several co-occurrences of taxon pairs within the studied groups. As expected, the HCs and individuals with UC exhibited greater enrichment (137 and 138 pairs, respectively), indicating greater homogeneity among the samples. In contrast, the CD group exhibited only 93 pairs. When analyzing the IBD cohort, which included both UC and CD specimens, only 25 pairs of taxa were retrieved, indicating a high level of heterogeneity among the pooled IBD specimens. A detailed analysis highlighted pairs with opposite values of co-occurrence compared to the HCs. In the CD patients, opposite co-occurrence values were observed in two pairs (13–63 and 37–70), and three were observed regarding UC (12–45, 12–63 and 13–51). These findings imply that the microbial community structure in both CD and UC is disrupted, which could contribute to disease pathology.

Pair 13–63, representing *Clostridium* and an unannotated genus from the *Clostridiales* order, could suggest a hypothesis: symbiont *Clostridiales* numbers decrease in CD, whereas the abundance of pathobionts from the *Clostridium* genus increases.

In the UC group, pair 13–51 exhibited a negative correlation coefficient, whereas it was positively correlated in the HCs. For this pair, one could hypothesize that the abundance of the symbiont *Anaerostipes* genus, a producer of butyric acid, decreases in UC, whereas the abundance of the pathobiont *Clostridium* genus increases. Both pairs 12–45 and 12–63 co-occurred with a positive correlation coefficient in the UC group, but this correlation coefficient was negative in the HCs. The main sources of energy for *Bacteroides* (12) species in the gut are complex host-derived and plant glycans and represent one of the main genera reduced in fecal samples of patients with CD [51]. The unannotated genus from the *Clostridiales* order (63) is classified as a symbiont, and feature 45 represents an unannotated genus from the *Barnesiellaceae* family known to produce elevated quantities of SCFAs.

The *Clostridiales* order (63) was also highlighted through the betweenness centrality local metric and keyplayers metrics as the most important node in the network of control subjects, although it also had one of the best scores in the UC cohort. However, in the UC group, the highest value was achieved by feature 35, namely the *Coprococcus* order, which also yielded a good score concerning fragmentation. *Coprococcus*, an order of butyrate-producing bacteria, was found to have significantly reduced abundance in patients with IBD, especially those with UC [52]. 

Furthermore, these analyses emphasize the role of the *Clostridium* genus (13), which, in our study, was found in greater abundance in IBD patients, indicating its potential involvement in this disease’s pathology. Additionally, our findings highlight the role of the *Slackia* genus (32) in CD patients. The *Slackia* genus may play an important role in the gut health of human beings [53], and it was previously found to be significantly abundant in UC patients who achieved clinical remission after undergoing a fecal microbiota transplantation (FMT) [54]. 

When we analyzed baseline microbial composition data to identify predictor(s) for responses to biologic agents, we found that IBD patients who responded to biologics at 14 weeks exhibited a higher abundance of *Odoribacter* (29) and *Ruminococcus* (52). Both taxa are known for their beneficial properties and commensal roles, primarily attributed to their ability to produce SCFAs. 

The existing literature data depict the *Odoribacter* genus as a cornerstone in generating protective immunity linked to TH17 immune responses [55]. Its beneficial functions encompass promoting healing during colitis, modulating regulatory T-cell responses and contributing to gut health. Lima and her collaborators [56] proposed that a strain within the *Odoribacter* genus, namely *Odoribacter* splanchnicus, is a key component promoting both metabolic and immune cell protection from colitis. 

On the other hand, the *Collinsella* genus (49) showed a higher prevalence among IBD patients who did not respond to biologics at 14 weeks. Recent evidence shows an increase in *Collinsella* numbers in active UC patients [57]. At 52 weeks, a genus identified as *SMB53* from the *Clostridiaceae* family (73) was more enriched in IBD patients who responded to biologic therapies. Although the function of *SMB53* is currently under debate, this genus has been associated with variants in the GNA12 gene, implicated in barrier defense and linked to UC [58]. 

It is important to note that the *Prevotella* genus was associated with a response to biologics at 14 and 52 weeks. However, this association was only observed with respect to Coda-lasso at 14 weeks and Coda-lasso and Clr-lasso at 52 weeks. The presence of the *Prevotella* genus was related to the presence of enterotype 2 (Bact2), which, in a recent study, was associated with significantly higher remission rates at baseline in patients hosting Bact2 (i.e., *Prevotella*) receiving anti-TNF therapy compared to those receiving Vedolizumab [59]. 

A greater abundance of an unclassified genus from the *Barnesiellaceae* family and one from *Lachnospiraceae*, both known to produce elevated quantities of SCFAs, was observed in IBD patients responding to Vedolizumab at 14 weeks. Regarding CD patients, a previous study linked an increase in *Lachnospiraceae* to treatment efficacy after 6 weeks of Infliximab therapy [60]. In addition, in this group of CD patients, the *SMB53* genus was found to be enriched, although significance was achieved for two of the three predictive tools used. 

The unclassified genus from the *Barnesiellaceae* family also exhibited increased abundance in UC patients responding to Vedolizumab at 14 weeks, whereas the *Collinsella* genus was significantly more present in UC non-responders. In addition, the *cc_115* genus from the *Erysipelotrichaceae* family showed greater abundance in UC non-responder patients using Vedolizumab at 52 weeks. The abundance of the *cc_115* genus was found to be decreased in CD patients at the time of surgery [61]. The *Erysipelotrichaceae* family is known to produce acetate, propionate and butyrate. These findings underscore the importance of a baseline presence of beneficial bacteria for overall well-being. 

When we compared the microbial compositions at baseline with those at 14 and 52 weeks of therapy, the data revealed that the *Enterococcus* genus was more abundant at baseline than after 1 year of biologic therapy. *Enterococcus*, representing a genus of opportunistic pathogens, may suggest a potential association with disease activity, given its enrichment at the baseline level. 

We acknowledge certain limitations of this study, including the relatively small number of enrolled patients (though this number was in line with the nature of a single-center series) and the restricted number of specimens available at 52 weeks. However, the heterogeneous ages and previous therapies limited our ability to reach generalizable conclusions. 

We encountered several technical challenges during this study. Initially, we generated a substantial number of high-quality reads, prompting the application of multiple rarefaction tests and feature/sample filtering criteria to ensure reliable diversity metrics and eliminate ultra-rare taxa. Additionally, we employed various compositional data analysis methods, including alternative approaches. Moreover, our computational approach enabled us to identify and quantify differences in microbial content among patient cohorts and control samples.

It is well established that the gut microbiome plays a pivotal role in the onset and progression of IBD. Through microbiome 16S data analysis, we strengthened the connection between IBD and microbial dysbiosis, although our cohort was composed of patients with a history of multiple different therapies and, at recruitment, did not have a de novo diagnosis. Among the various forms of IBD, CD was associated with a higher degree of dysbiosis compared with UC, as supported by alpha and beta diversity metrics and microbial ecosystem analysis using the Lloyd–Price test. 

Patients with a pronounced degree of dysbiosis at baseline were less likely to respond favorably to biologic therapy. Subsequent analyses employing compositional data analysis methods highlighted the absence or depletion of key taxa responsible for producing SCFAs and identified an excess of pathogenic or pathobiont taxa. When we dissected the co-occurrence network of the bacterial community, dense and complex networks were detected in the HC and UC patients, whereas sparse networks were found in the CD and IBD patients. Many correlation coefficients between two taxa changed from a direct positive co-occurrent correlation to a negative one when comparing the HCs with the CD or UC patients, indicating the depletion of beneficial bacteria or, conversely, the proliferation of pathobionts. 

## 5. Conclusions

Patients with IBD, especially those with CD, have an indigenous dysbiotic microbial composition. We identified microbial signatures associated with a response or resistance to biologics. Microbial variation, with respect to specific commensal and butyrate-producing bacteria, as well as pro-inflammatory gut bacteria, may determine whether a patient responds to biologic therapy. Understanding these microbial dynamics could pave the way for tailored therapeutic interventions aimed at modulating the microbiota for sustained remission.

## Figures and Tables

**Figure 1 microorganisms-12-01260-f001:**
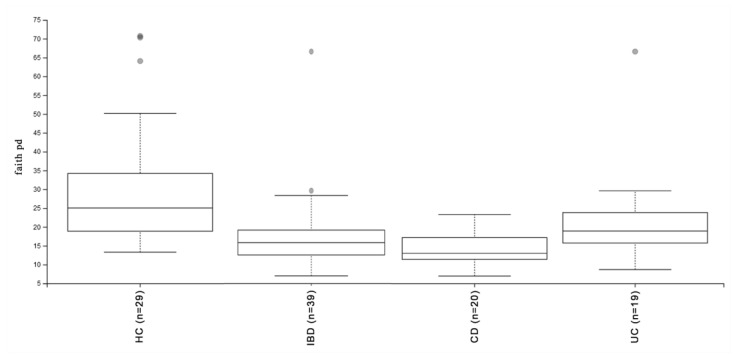
Boxplot for group-specific Faith’s phylogenetic diversity values (see Table 2 and Appendix A for details). The Kruskal–Wallis test for all groups had a q value < 0.015. IBD: inflammatory bowel disease; CD: Crohn’s disease; UC: ulcerative colitis; HC: healthy controls.

**Figure 2 microorganisms-12-01260-f002:**
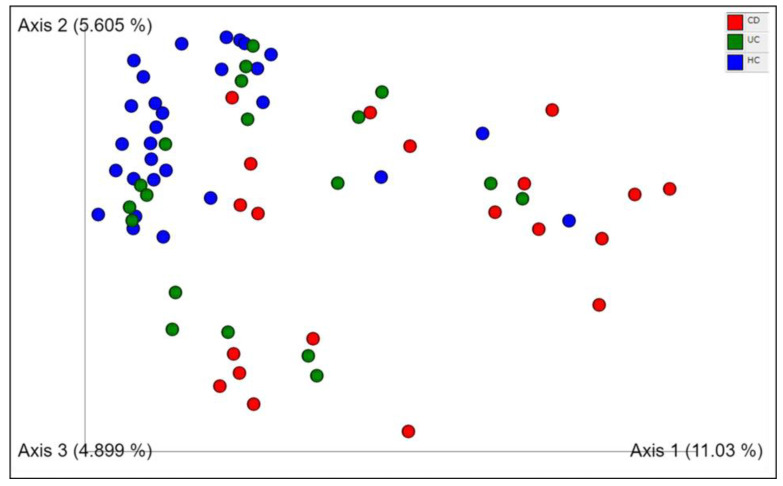
Principal coordinates analysis plot for Crohn’s disease (red), ulcerative colitis (green) and healthy control (blue) sample groups; distances across samples were calculated using the Bray–Curtis index. More details are available in Appendix A. CD: Crohn’s disease; UC: ulcerative colitis; HC: healthy control.

**Figure 3 microorganisms-12-01260-f003:**
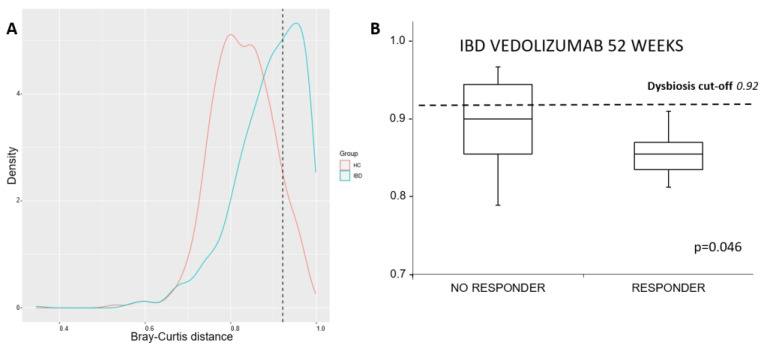
The degrees of dysbiosis among all the individuals with IBD at baseline and HC samples were assessed using the Lloyd–Price test. The blue line represents IBD patients, and the red line represents HCs; (panel **A**) Box plot of IBD patients who responded and those who did not respond according to their degree of dysbiosis at baseline (panel **B**); dysbiosis cut-off = 0.92. More details are available in Appendix A. IBD: inflammatory bowel disease; HC: healthy control.

**Table 1 microorganisms-12-01260-t001:** Clinical features of individuals with Crohn’s disease and ulcerative colitis. E1: proctitis; E2: left colitis; E3: extensive colitis; L1: ileal; L2: colonic; L3: ileocolonic; L4: upper gastrointestinal tract; B1: inflammatory behavior; B2: stricturing behavior; B3: penetrating behavior.

Characteristics	n [%] or Mean, and Range
No. of patients	39
Female	14 [35.9%]
Crohn’s disease [CD]	20 [51.3%]
Ulcerative colitis [UC]	19 [48.7%]
Montreal classification, UC	19
E1/E2/E3	2 [10.5%]/10 [52.6%]/7 [36.8%]
Montreal classification, CD	20
L1/L2/L3/L4	13 [65%]/1 [5%]/6 [30%]/0 [0%]
B1/B2/B3	8 [40%]/4 [20%]/8 [40%]
Perianal disease	4 [20%]
Mayo score at baseline [UC patients]	9.0
Harvey–Bradshaw Index (HBI) score at baseline [CD patients]	8.5
Age at diagnosis, years	34.5 [9–65]
Duration of disease, years	12.3 [4–42]
Age at baseline, years	42.56 [15–72]
Smoking	9 [23.1%]
Extraintestinal manifestations	
Arthritis/sacro-ileitis	12 [30.8%]
Skin manifestations	4 [10.3%]
Iritis/uveitis	1 [2.6%]
Primary sclerosing cholangitis	0
Previous surgery	8 [20.5%]
Previous biologic therapy	14 [36.9%]

**Table 2 microorganisms-12-01260-t002:** Summary of pairwise group comparisons for four alpha diversity indices. Benjamini and Hochberg corrected *p*-values (q-values) for Kruskal–Wallis tests are shown. IBD: inflammatory bowel disease; CD: Crohn’s disease; UC: ulcerative colitis; HC: healthy control.

	HC vs. IBD	HC vs. CD	HC vs. UC	UC vs. CD
Pielou’s evenness	0.0019	0.00014	0.20	0.0030
Faith’s phylogenetic distance	0.000017	0.000003	0.015	0.013
Number of observed features	0.00000015	0.00000019	0.0007	0.0007
Shannon’s entropy	0.000002	0.000002	0.003	0.00015

**Table 3 microorganisms-12-01260-t003:** Summary of pairwise group comparisons for beta diversity measures. Benjamini and Hochberg corrected *p*-values (q-values) for PERMANOVA tests are shown. IBD: inflammatory bowel disease; CD: Crohn’s disease; UC: ulcerative colitis; HC: healthy control.

	HC vs. IBD	HC vs. CD	HC vs. UC	UC vs. CD
Bray–Curtis dissimilarity	0.00015	0.00020	0.00020	0.00030
Jaccard similarity	0.00015	0.00020	0.00020	0.00024
Unweighted UniFrac dissimilarity	0.00015	0.00020	0.00020	0.0067
Weighted UniFrac dissimilarity	0.0018	0.0012	0.049	0.26

**Table 4 microorganisms-12-01260-t004:** Number of features and links in each network and in random graphs with the same number of nodes at different probabilities of connection (25%, 50% and 75%), also revealing how the increase in the number of links increases the density of the network. The last column contains the maximum number of links allowed for the given number of features. Random graphs were generated using the Erdős–Rényi model. IBD: inflammatory bowel disease; CD: Crohn’s disease; UC: ulcerative colitis; HC: healthy control, p: probability of connection.

Network	Features (Nodes)	Links/Density	*p* = 0.25 Links/Density	*p* = 0.50 Links/Density	*p* = 0.75 Links/Density	Max Links
IBD	23	25/0.099	56/0.221	130/0.514	192/0.76	253
CD	74	93/0.034	710/0.262	1351/0.5	1958/0.725	2701
UC	78	138/0.046	773/0.257	1545/0.514	2245/0.748	3003
HC	57	137/0.086	408/0.256	750/0.47	1214/0.76	1596

## Data Availability

The original contributions presented in the study are included in the article/Appendix A, further inquiries can be directed to the corresponding author.

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
