# Peer review of "Deciphering Microbial Composition in Patients with Inflammatory Bowel Disease: Implications for Therapeutic Response to Biologic Agents"

_microorganisms, 2024, doi:10.3390/microorganisms12071260_

Round 1

Reviewer 1 Report (Previous Reviewer 4)

Comments and Suggestions for Authors

The Authors revised and corrected the manuscript according to my suggestions/comments. Now, it appears clear, tidy and professional. I support its publication.

Comments on the Quality of English Language

Generally good. Only a few typos.

Reviewer 2 Report (Previous Reviewer 5)

Comments and Suggestions for Authors

Rewriting of the Discussion and Conclusion Sections: The revisions made to the discussion and conclusion sections have addressed the previous concerns about readability. The authors have incorporated changes that clarify the impact of heterogeneous ages and previous therapies on the generalisability of the study’s findings. This acknowledgment enhances the manuscript by providing a more realistic interpretation of the results.

English Language and Grammar: The manuscript has seen improvements in language use and grammar. The authors have had the text reviewed by experts from the MDPI English service, as mentioned. These revisions have enhanced the readability of the manuscript.

Clarification of the Study's Limitations: The authors have acknowledged the limitations concerning the small sample size and the diversity of therapies received by the study participants. 

These adjustments have improved the manuscript, making the study's context and findings clearer and more reliable.

This manuscript is a resubmission of an earlier submission. The following is a list of the peer review reports and author responses from that submission.

Round 1

Reviewer 1 Report

Comments and Suggestions for Authors

Orazio Palmiersi et al. provides valuable insights into the link between gut microbiome alterations (dysbiosis) and the development and treatment of Inflammatory Bowel Diseases (IBD), such as Crohn's disease (CD) and ulcerative colitis (UC).  Recent research has focused on understanding the role of the gut microbiome in IBD and its implications for treatment with biologic agents. These biologic therapies target specific molecules involved in the inflammatory response, aiming to reduce inflammation and manage symptoms in IBD patients.

There are a few minor points that should be brought to the attention of the authors:

1.      There is no need to highlight headings in the abstract: methods, results, etc.

2.      Keywords should be listed alphabetically.

3.      The characteristics of the study group are insufficient. Consider adding a table with demographic and clinical parameters of IBD patients and healthy subjects.

4.      There are no inclusion criteria for the study.

5.      All abbreviations in the table should be explained below the tables.

6.      The literature record should be corrected according to the guidelines.

Comments on the Quality of English Language

Minor editing of English language required

Author Response

We appreciate the reviewer's comments and have addressed all points accordingly. The feedback provided has significantly contributed to the improvement of our manuscript in its revised version.

Specifically, we added the inclusion criteria for both patients and controls, included abbreviations in the tables, and corrected the references as necessary. We added further characteristics in Table 1 " Clinical features of individuals with Crohn's disease and ulcerative colitis ". Expert colleagues from the MDPI English service carried out a full review of the English version.

Reviewer 2 Report

Comments and Suggestions for Authors

It is an interesting work basically trying to explain why some IBD patients fail to achieve or to maintain disease remission.

I have some questions:

1.      Lines 89-92 “66 stool samples from 39 IBD patients and 29 HCs. Among IBD patients starting biologic therapies, 39 specimens were collected at baseline, 11 at 14 weeks, and 16 at 52 weeks.” Then, Lines 418-420 “Of the 39 IBD patients treated with biologics, 22 achieved clinical remission at 14 weeks”, and Lines 427 “The IBD patients responding to Vedolizumab at 14 weeks had…” How many were the patients [out of 39] treated with biologics? To conclude to 22 patients with remission? And why the stool samples on week 14 was only 11? On how many samples did the analysis performed?  I would like please a table analyzing the number of patients in each group and subgroup throughout the study.

2.      Table 2: There is no data for patients age at baseline. However, you referred “age of diagnosis 34.5 years and a range between 9 and 65years.” And then you referred to disease duration 12.3years, range 4 to 42 years. Hypothetically saying, the oldest patient had diagnosed at 65 and had the shorter duration of treatment, 4 years; this means he/she was 69 upon the entry to the study. Again. The younger being 9 years old at diagnosis and being 4 years only in treatment, he/she is 13 years upon the entry to the study. I wonder if this difference in age 13 and 69 pose a bias in your results, being well known the change in the microbiome during the progress of life. I would like to be commented in the limitations of the study, across with the small number of different status patients.

3.      conclusions: the fecal microbiota transplantation in capsules is something new, mainly applied to C. difficile infections. To my opinion, this future therapy must not used as a concluding sentence, presented thus as superior to probiotics.

Author Response

We appreciated the reviewers' comments, which have improved the manuscript. We have considered all the points and corrected them accordingly.

In particular, we added a table in Supplementary File 1, titled "1.2.1 - Number of patients analyzed in each group and subgroup throughout the study," and addressed the issues raised in point #2 within the limitations section of the study. Additionally, we have rewritten the conclusion as suggested, removing the sentence regarding FMT capsule-based therapy. Furthermore, a full review of the English version was carried out by expert colleagues from the MDPI English editing service.

Reviewer 3 Report

Comments and Suggestions for Authors

This study lacks a hypothesis. In other words, what each analysis is aiming to reveal is unclear. Results from each experiment stand alone and not logically connected to clarify the relationship between microbiome and resistance to the therapy. In addition, experimental condition, each figure including its resolution are very hard to be understood objectively, that also make this study difficult to be understood.

Besides, analysis of microbiome likely to show non-specific significant differences between experimental groups statistically due to the comparison of many multiple elements. Interpreting the meaning afterward is not acceptable in science. In addition, analysis of microbiome includes very abstract data as like figure 4, that cannot be evaluated whether physiologically significant correlations between microbiome and resistance to therapy are observed or not. If the authors claim that the evaluation succeeded to achieved, what difference of microbiome is responsible for the effectiveness of the therapy should be clearly described. Generally, it’s impossible.

Together with these things, it's necessary to reconsider the appropriate methods and procedures from scratch to achieve what the authors want to elucidate. Clear explanation of the results to readers is also essential.

Comments on the Quality of English Language

Way of presentation should be improved to make the contents understood by readers of the manuscript.

Author Response

We apologize to the reviewer for the lack of clarity throughout the text or the poor quality of the figures they found in the paper. In the revised version, we have included comments from additional reviewers, which improved the manuscript.

We have enhanced the figures in the main text and decided to delete Figure 4. Furthermore, a full review of the English version was carried out by expert colleagues from the MDPI English editing service.

Reviewer 4 Report

Comments and Suggestions for Authors

Without any doubt, there was a lot of work included in this very complex research. The manuscript contains a lot of analyses and comparisons, made through various methods. The main limitation (also written by the Authors, but still a limitation) is the small number of patients, as only those with Vedolizumab were analyzed in the end (25 patients). Also, an important limitation is considering nowadays, just clinical remission, and not also “mucosal healing”, or at least, some surrogates – like “fecal calprotectin”. Not even inflammatory serum markers were considered. I suggest rewriting the manuscript, with these data included and analyzed. Another remark: The aim I understand was to “identify microbial signatures that predict treatment response and improve clinical management” to biologics. However, the authors did a lot of research on dysbiosis in CD and UC patients, comparing them with healthy controls, at baseline (and also commented on this dysbiosis in ‘Discussion”). But this has no point since IBD patients were NOT “de novo diagnosed”, but had a long history of the disease (mean 12.3 years), during which they probably had different therapies, including surgery. This is not like they were at the onset, before any therapy given.

I have listed other comments/questions for consideration, below:

1.       Title: would the Authors kindly consider changing from “dissecting” to “deciphering”?

 2.       Abstract:

a.       Introduction: “While IBD often requires the use of immunosuppressant drugs and biological therapies for clinical remission, many patients do not benefit from these drugs, and the reasons remain poorly understood.” First, please correct – it is not only clinical remission, but also mucosal healing. Second, please revise: Most patients benefit from immunomodulators and biologics.

b.       Instead of “This study focuses”, I suggest inserting a proper aim.

c.        Methods: Please mention how many with CD and UC, separately. I would also suggest to make it clear that it was a prospective study, as it was mentioned in the main manuscript. This aspect increases its value and it should be included in the Abstract as well.

d.       Please mention that this was before starting biologics: “Differences were observed in alpha and beta metrics among patients with CD, UC, and 26 HC, as well as between CD and UC groups.”

e.       Conclusion: “Our analyses identified an abundance of pathobionts microbes in IBD patients particularly among non-responders to biological therapies.” In what way? Only Collinsella were more abundant in those patients with non-response.

 3.       Introduction:

a.       There are many references about increasing prevalence published more recently. Please update.

b.       References [2] and [3] about genetics are old. Please update.

c.        Line 52: Please read recent literature – in IBD, the main target is nowadays the mucosal healing towards achieving deep remission, not the absence of symptoms (clinical remission) and update text and references. Please correct about clinical remission everywhere in the main text, according to STRIDE-II and most recent guidelines.

d.       All references are old (except for reference 15). Please update.

e.       Please also delete redundant sentences (with the same meaning).

f.         Aim should be rephrased.

 4.       Materials and Methods

a.       Please write how many patients with UC and CD. Please include more in “exclusion criteria for HC” (what about probiotics, synbiotics, prebiotics, postbiotics, other medication used – even natural supplements, vitamins etc etc). In fact,, how were HC chosen? This is very important, to avoid biases.

b.       Generally, there are too few samples during follow-up.

c.        Disease activity – just clinical scores. Not even fecal calprotectin or serum biological markers.

d.       Why were comparisons made for Vedolizumab only? (I found the reason only by the end of Results…but data were not shown).

 5.       Results

a.       It appears that 19 had UC and 21 had CD (“Among UC patients, 2 had proctitis, 10 had distal colitis, and 7 had pancolitis. Disease localization in CD patients was: 13 ileitis, 1 colitis, and 7 ileocolitis.). That sums up 40 patients, not 39. In Table 1, the number of CD is 20. Please correct and clarify. In Table 1, L3 represents 6 patients, not seven.

b.       Also, none of CD patients had L4?

c.        Also, these patients had previous medication, that could impact on gut microbiota, including surgery (“mean duration of disease was 12.3 years”)

d.       What was the reason for including the only patient on “Ustekinumab”, according to the aim? Please clarify.

 6.       Discussion

a.       I suggest starting with your own results and commenting on them/comparing to the new existing literature

b.       Please remove the sentence about antibiotics (reference [39]) – dated 2011, as they are not any more an option.

c.        I suggest in this paragraph to refer to  ”Vedolizumab”, when talking about biologics, as regarding anti-TNF agents – the authors mentioned: “Due to the limited number of subjects receiving anti-TNF therapy, no significant associations were identified in any of the comparisons (data not shown).

d.       Please replace ref. [40] with something recent. The same applies for other older references in this paragraph, while newer and better data were published.

e.       A proper discussion about future research would be excellent here and not in Conclusion.

f.         As I mentioned before, results and discussion about dysbiosis in IBD patients after so many years do not give us any clue.

7.       Conclusion should be rewritten and focus on the own results and proper aim.

Comments on the Quality of English Language

Should be markedly improved:

Comma use, noun-verb agreement, verb use, misspelled words, etc.

Author Response

We appreciated the reviewer’s comments that helped improve the manuscript. We concur with the reviewer's assessment of the limitations of our work. It should be noted that the patients included in the study were recruited from a single center and were not part of a clinical trial; rather, they were drawn from the context of daily clinical practice and in real-life settings.

While it is true that these studies have several disadvantages, they are essential for understanding how treatments and interventions work in everyday clinical practice, where patients have numerous challenges in arranging visits, blood tests, and instrumental exams.

  1. We have changed the title from “dissecting” to “deciphering”. This is an acceptable change.
  2. Abstract: All issues were assessed and corrected following the reviewer’s comments/questions. In particular, modifications were made to the sentences in the conclusion session.
  3. Introduction: When possible, the bibliography was updated; however, several milestone papers, particularly in the field of bioinformatics, could not be replaced. Additionally, we added a reference to the STRIDE-II study.

As suggested, we attempted to eliminate redundant sentences and rephrase the aim.

  1. Materials and Methods:

In this section, we added the inclusion criteria for both patients and healthy controls. We concur with the reviewer's comments regarding the sample size and surrogate markers. However, it should be noted that the patients were recruited from a single center and were not part of a clinical trial.

  1. We thank the reviewer for their attention. The L3 group represents six patients, not seven, and none of the patients included in the study had L4. Furthermore, data of previous biological therapy are included in Table 1. The sole patient who received Ustekinumab was included in the case-control study and the analyses of biologics.
  2. Discussion: All issues were appropriately addressed and rectified. The conclusion section was also revised.

Furthermore, the text was subjected to a comprehensive review by expert colleagues from the MDPI English service.

Reviewer 5 Report

Comments and Suggestions for Authors

This is an interesting investigation into a hypothesis which has shown some promise in early research. The authors have used appropriate methods and described them well. I am a clinical gastroenterologist and not a lab scientist with experience in the techniques used and I would strongly suggest the Editor seeks a review from someone with a lab focus in the techniques used. 

The manuscript is well written and the English language is, on the whole, very readable with some minor errors of grammar and syntax. However, the final discussion is very difficult to read and requires rewriting.

The outcome measure for comparison of clinical response across groups were weak in that no biological markers of disease activity were used, nor was endoscopic outcome available. This limits the conclusions that can be drawn from the data and runs the risk of confounding by non-inflammatory symptoms being associated with microbiome related outcomes. This weakness was not clearly addressed in the discussion.

The main weakness of this study is the very small number of patients available for follow up testing. Previous exploratory studies in the same area have included follow up samples of at least 3 times, and up to several times, the number of subjects. This is compounded by the inclusion of patients receiving multiple different therapies, where past studies have tended to focus on a single therapy. While these weaknesses are stated they do markedly limit the authors' ability to reach generalisable conclusions from their data.

The discussion is generally well written and does address these concerns, however, the final conclusion section is very difficult to read and draws conclusions that bear no relation to the data presented and cannot be substantiated by the results. This requires a complete rewrite.

Comments on the Quality of English Language

Generally good, some proof reading required. Conclusion needs complete rewrite.

Author Response

We appreciated the reviewer’s comment that improved the manuscript. We accounted for all the issues and corrected them accordingly.

In particular, the discussion was revised, and the conclusion rewritten, with the main text noting that "the heterogeneous ages and previous therapies limit the ability to reach generalizable conclusions."

In addition, a full review of the English text was conducted by expert colleagues from the MDPI English service. 

Round 2

Reviewer 2 Report

Comments and Suggestions for Authors

Dear authors

thanks you for making the changes I have suggested

One more comment on the Suppl Table 1.2.1: It is better to arrange sub-groups in a different way to be clear that all your patients were 39; subjected to biologic treatment 38 [studied in 14 and again in 52wks] ; and these 38 subdivided into Vedolizumab or anti-TNF [studied in 14 and again in 52wks]

Author Response

We appreciate the reviewer's comment and agree that further clarification in the description of Supplementary Table 1.2.1 was needed. We have modified the table accordingly.

Reviewer 3 Report

Comments and Suggestions for Authors

The editor sent revised version despite of my previous reject decision.

Author Response

N/A

Reviewer 4 Report

Comments and Suggestions for Authors

The manuscript was improved, but not in essential points.

Main comments:

A. The authors wrote in the Abstract: “The aim of this study was to investigate the gut microbiome of IBD patients using biologics to identify microbial signatures associated with response”.

1st: Please specify what type of response (clinical is not enough).

2nd: Conclusion has to refer to the aim. However, conclusion does not show anything associated with response, but with non-response (in the Abstract).

B. Material and methods: The Authors corrected: “Response to therapy was assessed using clinical indices (HBI <5 and PMS <2) and at least one objective marker of disease activity (a CRP level < 5mg/dl, a fecal calprotectin level < 150 µg/g, a reduction of at least 3 points in SES-CD or ≥ 1 point in endoscopic Mayo score, or a reduction of at least 2 mm in bowel thickness as assessed by bowel ultrasound or small-bowel MR enterography)”.

However, later, in “2.1. Assessment of disease activity” – only clinical scores were written. Please clarify and correct.

C. Also, later, in the whole manuscript, everything refers to clinical response. Corrections are needed. Serum inflammatory markers and fecal calprotectin have to be part of the daily basis assessment, nowadays. It is not about trials. It is about offering our patients the best of care. It is unfair for patients to have GIs that consider only clinical data. This would jeopardize the course of their disease. Therefore, the results of this study do not help our patients. Tight control is needed.

Thank you

Comments on the Quality of English Language

Overall, improved. Some typos have to be corrected.

Author Response

We are grateful for the insightful feedback provided by the reviewer, which has given us the opportunity to better clarify some areas of improvement.

  1. A) Abstract: As requested, we have modified the sentence as follows:

 “The aim of this study was to investigate the gut microbiome of IBD patients using biologics to identify microbial signatures associated with response, following standard accepted criteria.”

Conclusions: As per your request, we have incorporated an additional sentence, presented as follows: “Furthermore, specific bacteria producing SCFAs were abundant in patients responding to biologics and in those responding to Vedolizumab”.

  1. B) We improved the assessment of the disease activity section by better specifying the meaning of responders to specific biologics according to the known established criteria. In particular, we have added “We considered responders to specific biologics as those patients showing a reduction of ≥2 points in PMS and ≥3 points in HBI from baseline, associated with at least one of the following objective markers of inflammation: a CRP level < 5 mg/dL, a fecal calprotectin level < 150 µg/g, a reduction of at least 3 points in SES-CD or ≥1 point in the endoscopic Mayo score, or a reduction of at least 2 mm in bowel thickness as assessed by bowel ultrasound or small-bowel MR enterography.”
  2. C) We agree with this reviewer; we have better defined the meaning of the response, and corrections have been made in the text for clinical response.

Moreover, a comprehensive review of the English manuscript was carried out again by skilled peers affiliated with the MDPI English editing service.